# A causality-based divide-and-conquer algorithm for nonequilibrium Green's function calculations with quantics tensor trains

Ken Inayoshi[1]$^\star$, Maksymilian Środa[2], Anna Kauch[3], Philipp Werner[2], and Hiroshi Shinaoka[1]

**1** Department of Physics, Saitama University, Saitama 338-8570, Japan
**2** Department of Physics, University of Fribourg, 1700 Fribourg, Switzerland
**3** Institute of Solid State Physics, TU Wien, 1040 Vienna, Austria

$\star$ kinayoshi@mail.saitama-u.ac.jp

## Abstract

We propose a causality-based divide-and-conquer algorithm for nonequilibrium Green's function calculations with quantics tensor trains. This algorithm enables stable and efficient extensions of the simulated time domain by exploiting the causality of Green's functions. We apply this approach within the framework of nonequilibrium dynamical mean-field theory to the simulation of quench dynamics in symmetry-broken phases, where long-time simulations are often required to capture slow relaxation dynamics. We demonstrate that our algorithm allows to extend the simulated time domain without a significant increase in the cost of storing the Green's function.

# 1  Introduction

The nonequilibrium Green's function (NEGF) method [1–5] is a versatile and widely used approach for investigating nonequilibrium phenomena in quantum many-body systems. This method enables the calculation of the time evolution of various physical observables, including the electron density, energy, order parameters, and single-particle spectra. The main computational bottleneck of the NEGF method is solving the nonequilibrium Dyson equation (the Kadanoff-Baym equation), which involves a convolution integral with respect to time. This requires a data size of $\mathcal{O}(N_t^2)$ for storing Green's function and a computational cost of $\mathcal{O}(N_t^3)$ to solve the Dyson equation, where $N_t$ denotes the total number of time steps [3, 6]. Due to this rapidly growing memory and computational cost, it is difficult to simulate nonequilibrium dynamics in large lattice systems up to long times.

Various techniques have been proposed to overcome this bottleneck. The generalized Kadanoff-Baym ansatz (GKBA) [7–33], based on the Hartree-Fock approximation, calculates the time evolution of the single-particle density matrix. This approximation drastically reduces the data size and computational complexity to $\mathcal{O}(N_t)$ and $\mathcal{O}(N_t^2)$, respectively. A clever reformulation of the GKBA, known as the G1-G2 scheme [20, 21, 28, 31], further reduces these complexities to $\mathcal{O}(1)$ for memory and $\mathcal{O}(N_t)$ for the computation. This memory reduction enables the use of the GKBA in various setups, including first-principles calculations of electron dynamics [12, 19, 30]. Although this method is powerful in the weak-coupling regime, its applicability to strongly correlated systems is nontrivial. Another widely used approach is the memory truncation method [34–38]. This method exploits the decay of the self-energy with respect to relative time and truncates the memory of the self-energy with a cutoff time. A study combining dynamical mean-field theory (DMFT) with this method showed that the order parameter can be evaluated up to times of $\mathcal{O}(1000)$ inverse hoppings [35]. However, the effectiveness of the truncation approach depends on the problem. For example, in a study of superconductivity within the fluctuation exchange approximation (FLEX) in three-dimensional systems [37, 39], the self-energy with momentum closest to the Fermi surface was shown to decay slowly with respect to relative time.

New promising avenues to alleviate the above problems focus on the intrinsic structure of the Green's functions and combine NEGF methods with efficient memory compression techniques. One of the pioneering approaches is the hierarchical off-diagonal low-rank method [40–42]. This method partitions the lower-triangular part of the Green's function into off-diagonal

block matrices and compresses them using the singular value decomposition (SVD). The computational and memory complexities are reduced by nearly one power of $N_t$. A recent study [42] demonstrated that this method enables simulation of a superconducting system up to sufficiently long times.

Another interesting memory compression technique is a tensor-network approach based on the so-called quantics tensor trains (QTT) [43–46]. In the QTT method, the time dependence of the Green's function is represented by a tensor train (TT), whose data size is significantly compressed owing to the empirical observation that different length scales are not strongly correlated with each other leading to low bond dimensions [45, 46]. Operations essential to the NEGF calculations, such as element-wise products and convolution integrals, can be efficiently performed in the compressed form [45]. In an early study combining QTT and NEGF methods [47], the Dyson equation was solved iteratively, and the benchmark tests were limited to short time intervals of $\mathcal{O}(1)$ inverse hopping times. However, in our recent study [48], we greatly improved the implementation, in particular, by introducing a linear equation solver to solve the nonequilibrium Dyson equation in a stable fashion. This allowed us to demonstrate nonequilibrium simulations up to a scale of $\mathcal{O}(100)$ inverse hopping times for a large two-dimensional lattice system with more than 4000 sites.

However, there is still room for improvement in the current QTT-based NEGF (QTT-NEGF) implementation. The main limitation is that this implementation does not exploit the causality of Green's functions. In conventional approaches [3, 6, 30], causality is utilized to gradually extend the maximum time $t_{\max}$ of the simulation, and to restrict the calculation of the Green's functions to the newly added time domain. In contrast, the current QTT-based calculation globally updates the Green's functions on the whole time domain within the self-consistent loop. This is not only computationally expensive, but also increases the number of iterations required for convergence in the limit of large $t_{\max}$, making it difficult to obtain a converged solution with high accuracy in a reasonable amount of time. Therefore, it is desirable to incorporate causality into the QTT-based method.

In this study, we propose a divide-and-conquer algorithm for QTT-NEGF calculations that exploits the causality of the Green's functions. We apply this method to analyses of symmetry-broken (antiferromagnetic) states using nonequilibrium DMFT [3, 49–54]. First, we show that the global update method, which does not exploit causality, becomes inefficient for long-time simulations, where many iterations are required to achieve convergence even when starting from a good initial guess close to the exact solution of the Dyson equation. Then, we introduce the divide-and-conquer method which gradually extends the time domain. We show that this approach improves the convergence of the Green's functions in self-consistent calculations, and allows to extend $t_{\max}$ without a significant increase in the data size of the stored functions.

This paper is organized as follows. In Sec. 2, we briefly review the QTT-NEGF method introduced in Refs. [47, 48]. In Sec. 3, we propose a causality-based divide-and-conquer algorithm. In Sec. 4, we introduce the time-dependent Hubbard model and explain the nonequilibrium DMFT formalism for the antiferromagnetic state. In Sec. 5, we present the results of numerical simulations. We first discuss how the Green's functions are updated over the whole time domain in the conventional QTT-NEGF method. Then, we discuss the efficient divide-and-conquer method. Section 6 provides a summary and outlook.

## 2   QTT-based nonequilibrium Green's function method

In this section, we briefly review the QTT-NEGF method introduced in Refs. [47, 48].

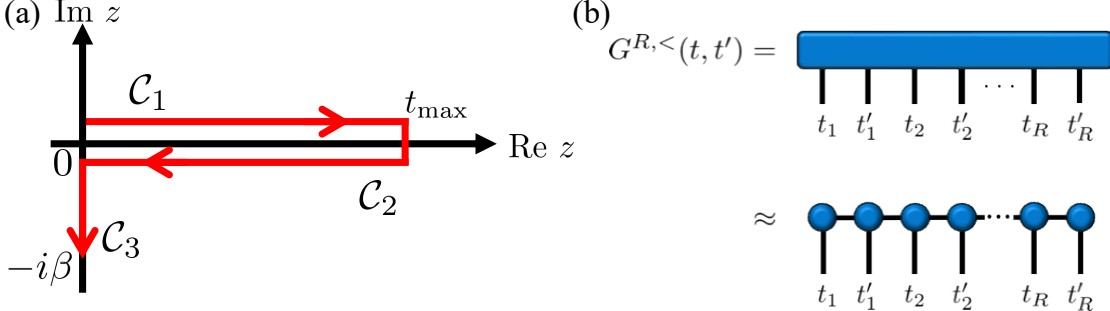

Figure 1: (a) Kadanoff-Baym contour, which consists of the forward real-time branch $\mathcal{C}_1$, the backward real-time branch $\mathcal{C}_2$, and the Matsubara imaginary-time branch $\mathcal{C}_3$. (b) QTT representations of the retarded ($R$) and lesser ($<$) Green's functions.

## 2.1 QTT representation of Green's functions

In the Kadanoff-Baym formalism, the contour-ordered Green's function $G(z, z')$ is defined on the L-shaped Kadanoff-Baym contour $\mathcal{C}$ [1,5], which consists of the forward real-time branch $\mathcal{C}_1$, the backward real-time branch $\mathcal{C}_2$, and the Matsubara imaginary-time branch $\mathcal{C}_3$ (Fig. 1(a)). Because the time variable $z$ belongs to three contours, $G(z, z')$ has $3 \times 3 = 9$ components. However, due to symmetry relations between the components [3], we only need to consider the Matsubara ($G^M$), retarded ($G^R$), left-mixing ($G^\rceil$), and lesser ($G^<$) components,

$$G^M(\tau, \tau') = G(-i\tau \in \mathcal{C}_3, -i\tau' \in \mathcal{C}_3), \tag{1}$$

$$G^R(t, t') = \theta(t, t') \left\{ G(t \in \mathcal{C}_2, t' \in \mathcal{C}_1) - G(t \in \mathcal{C}_1, t' \in \mathcal{C}_2) \right\}, \tag{2}$$

$$G^\rceil(t, \tau) = G(t \in \mathcal{C}_1, -i\tau \in \mathcal{C}_3), \tag{3}$$

$$G^<(t, t') = G(t \in \mathcal{C}_1, t' \in \mathcal{C}_2), \tag{4}$$

where $\theta(t, t')$ is the contour Heaviside step function and we define the Matsubara component with an extra $i$ factor over the standard convention, i.e., $G^M(\tau, \tau') = -iG(-i\tau \in \mathcal{C}_3, -i\tau' \in \mathcal{C}_3)$ [48]. In the QTT representation of a Green's function component, we first discretize each time axis using $2^R$ equidistant grid points, where $R$ is the number of binary digits (qubits) per time axis. The time indices are then enumerated from 0 to $2^R - 1$ and represented in binary form: $(z_1, z_2, \ldots, z_R)_2$ and $(z'_1, z'_2, \ldots, z'_R)_2$ for $z$ and $z'$, respectively, where each $z_r, z'_r \in \{0, 1\}$. Let us consider real times $t$ and $t'$ as an example. In this case, $(t_1, t_2, \ldots, t_R) = (0, 0, \ldots, 0)$ corresponds to $t = 0$, while $(t_1, t_2, \ldots, t_R) = (1, 1, \ldots, 1)$ corresponds to $t = t_{\max}$ [48]. We refer the reader to Refs. [45, 46, 48] for more detailed explanations.

As shown in Fig. 1(b), we arrange the discretized Green's function data as a $2R$-way tensor, and approximate it as a tensor train (TT) (or a matrix product state), i.e., a product of $2R$ three-way tensors, per each component. We use the interleaved representation [45, 46], where the binary digits corresponding to the same time resolution (e.g., $t_1$ and $t'_1$) are grouped next to each other in the tensor train. This arrangement exploits the expected strong entanglement between time indices at the same resolution scale [45], resulting in efficient compression. The data size of the resulting QTT is $\mathcal{O}(4RD^2)$, where $D$ is the maximum bond dimension. When $D \ll 2^{2R}$, the data size of the original Green's function is highly compressed. Previous studies revealed that equilibrium and nonequilibrium Green's functions are highly compressible in many cases [45, 47, 55–59]. As a simple example, let us note that the *non-interacting* Green's function can be represented as a QTT with very small bond dimensions, $D = \mathcal{O}(1)$ [45, 48]. This is because it is expressible by a combination of exponential and step functions, which factorize very well within the quantics representation [45].

## 2.2    NEGF calculations with QTT

Essential operations in NEGF calculations, such as summations, element-wise products, and convolution integrals, can be performed efficiently in the QTT representation [45, 47].

For instance, adding two TTs of bond dimension $D$ can be performed with the direct-sum algorithm and an SVD compression of the result to bring the bond dimension back to $\sim D$. This requires $\mathcal{O}(D^3)$ operations. While the density-matrix algorithm is in general more efficient, we do not use it here as it leads to a loss of accuracy (see the detailed discussion in Appendix A).

Next, the element-wise product of two TTs $A(z) = \Pi_i A_i(z_i)$ and $B(z) = \Pi_i B_i(z_i')$ ($A_i(z_i)$ and $B_i(z_i)$ are the $i$th core tensors of $A$ and $B$, respectively) can be expressed as the following contraction [45],

$$\sum_{z_1', \cdots, z_R'} \tilde{A}(z_1, z_1', \cdots, z_R, z_R') B(z_1', \cdots, z_R'), \tag{5}$$

where $\tilde{A}$ is given by

$$\tilde{A}(z_1, z_1', \cdots, z_R, z_R') = \left\{ A_1(z_1) \delta_{z_1, z_1'} \right\} \cdots \left\{ A_R(z_R) \delta_{z_R, z_R'} \right\}. \tag{6}$$

Assuming two TTs of bond dimension $D$ and a subsequent recompression to bond dimension $D$, the above requires $\mathcal{O}(D^4)$ operations with the fitting algorithm [47, 60].

The convolution integral can be numerically performed as a matrix multiplication [48]. Using the trapezoidal rule,

$$\int_0^{t_{\max}} d\bar{t} \, A(t, \bar{t}) B(\bar{t}, t') = \sum_{\bar{t}} A_{t\bar{t}} w_{\bar{t}} B_{\bar{t}t'}, \tag{7}$$

$$-i \int_0^{\beta} d\bar{\tau} \, A(\tau, \bar{\tau}) B(\bar{\tau}, \tau') = \sum_{\bar{\tau}} A_{\tau\bar{\tau}} v_{\bar{\tau}} B_{\bar{\tau}\tau'} \tag{8}$$

with the diagonal matrices $w = \mathrm{diag}(h_t/2, h_t, \cdots, h_t, h_t/2)$ and $v = \mathrm{diag}(-ih_\tau/2, -ih_\tau, \cdots, -ih_\tau, -ih_\tau/2)$. Here, $h_t$ and $h_\tau$ are the real and imaginary time steps, respectively. The QTT approach allows us to use very small values for $h_t$ and $h_\tau$, of the order of $< 10^{-6}$, leading to highly accurate integrals even with the simple trapezoidal rule. Note that the matrices $w$ and $v$ can also be expressed in the QTT representation with a maximum bond dimension $D = \mathcal{O}(1)$ [48]. For simplicity, we consider here the matrix multiplication of two TTs $A(z, \bar{z})$ and $B(\bar{z}, z')$, which can be expressed as the contraction [45],

$$\sum_{\bar{z}_1, \bar{z}_1', \cdots, \bar{z}_R, \bar{z}_R'} \tilde{A}(z_1, z_1', \bar{z}_1, \bar{z}_1', \cdots, z_R, z_R', \bar{z}_R, \bar{z}_R') B(\bar{z}_1, \bar{z}_1', \cdots, \bar{z}_R, \bar{z}_R'). \tag{9}$$

Similar to the element-wise product, assuming two TTs of bond dimension $D$, the above contraction and the subsequent recompression require $\mathcal{O}(D^4)$ operations with the fitting algorithm [47, 60].

## 2.3    Linear equation solver for the Dyson equation

In the previous study [48], we introduced a linear equation solver in which the nonequilibrium Dyson equation is expressed as a linear equation, $AX = b$, i.e., $[1 - G_0 * \Sigma *] G = G_0$ ($*$ denotes the convolution integral on the contour $\mathcal{C}$). Such a linear problem can be solved by a variational method similar to the density matrix renormalization group (DMRG) method [61, 62]. Our implementation of the solver is based on the tensor-network library ITensors.jl [63].

Using the integral formulation introduced in the previous subsection, we can write the Kadanoff-Baym equations for each physical component as

$$A^{\mathrm{it}}G^M = b^M, \tag{10}$$

$$A^{\mathrm{rt}}G^R = b^R, \tag{11}$$

$$A^{\mathrm{rt}}G^{\rceil} = b^{\rceil}, \tag{12}$$

$$A^{\mathrm{rt}}G^{<} = b^{<}. \tag{13}$$

Here, the linear operators $A^{\mathrm{it,rt}}$ and constant terms $b^{M,R,\rceil,<}$ are written as

$$A^{\mathrm{it}} = 1 - G_0^M v \Sigma^M v, \tag{14}$$

$$A^{\mathrm{rt}} = 1 - G_0^R w \Sigma^R w, \tag{15}$$

$$b^M = G_0^M, \tag{16}$$

$$b^R = G_0^R, \tag{17}$$

$$b^{\rceil} = G_0^{\rceil} + G_0^R w \Sigma^{\rceil} v G^M + G_0^{\rceil} v \Sigma^M v G^M, \tag{18}$$

$$b^{<} = G_0^{<} + G_0^R w \Sigma^{<} w G^A + G_0^{\rceil} v \Sigma^{\lceil} w G^A + G_0^{<} w \Sigma^A w G^A + G_0^R w \Sigma^{\rceil} v G^{\lceil} + G_0^{\rceil} v \Sigma^M v G^{\lceil}. \tag{19}$$

Below, we summarize the self-consistent calculation using the linear equation solver.

1. Prepare the initial guess for $G$ such as the non-interacting Green's function. The latter can be directly prepared as a QTT with small bond dimension $D = \mathcal{O}(1)$.

2. Using $G$, calculate the self-energy $\Sigma$ with the element-wise products and convolution integrals.

3. Using $G$ and $\Sigma$, solve the linear equation $A^{\mathrm{it,rt}}G^{M,R,\rceil,<} = b^{M,R,\rceil,<}$.

4. Get the new Green's function.

We repeat the above processes until convergence. Note that the linear equation for the Matsubara Green's function, which determines the equilibrium state, is decoupled from the other three equations. Therefore, we can first converge the Matsubara Green's function $G^M$ according to the above four steps. After $G^M$ is converged, the remaining three equations are solved sequentially in the order of Eqs. (11)-(13) at each iteration of the self-consistent loop [48].

## 3 Block time stepping based on divide-and-conquer algorithms

In this section, we propose a QTT-based divide-and-conquer algorithm to extend the time domain in a stable manner by exploiting the causality of the Green's functions. The flow of the divide-and-conquer algorithm is shown in Fig. 2.

### 3.1 Block structure for the real-time axes

Before explaining the divide-and-conquer algorithm in detail, we explain how to partition the time domain into two blocks along each real-time axis. The blocks are defined in Fig. 3. The first block along each real-time axis represents the past time domain ($0 \le t, t' \le t_{\max}$), and the second block represents the new time domain ($t_{\max} < t, t' \le 2t_{\max}$). QTTs for Green's functions and self-energies are defined over the entire time domain from 0 to $2t_{\max}$, but each QTT is projected on one particular block, i.e, set to zero outside that block. This means that, for example, the retarded component is actually represented by four QTTs (Fig. 3(a)). Since

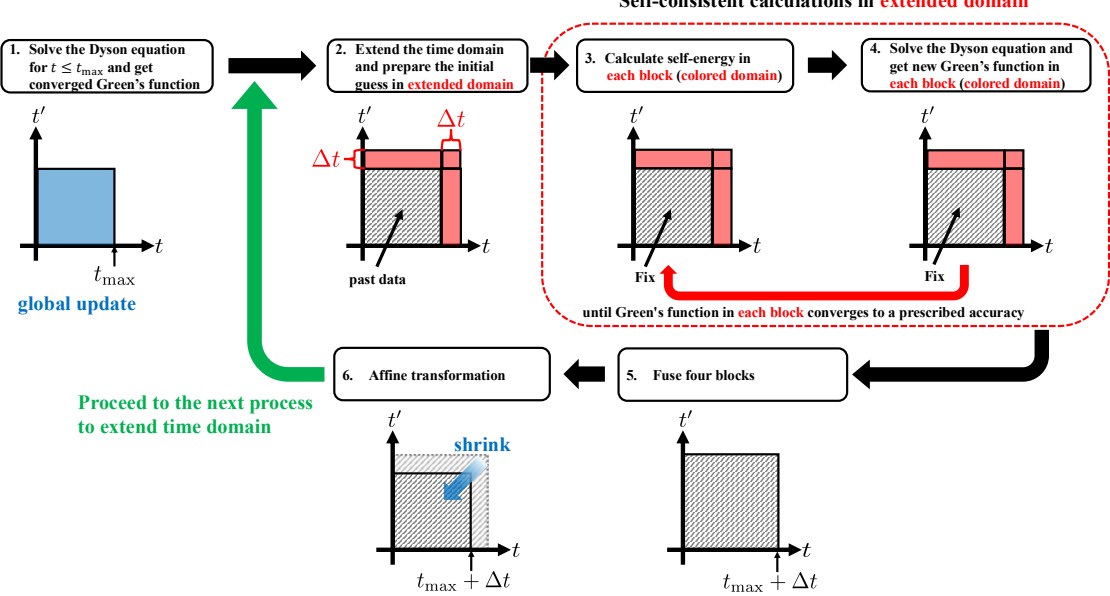

Figure 2: Flow of the divide-and-conquer algorithm. 1. Solve the Dyson equation with the linear equation solver for $t \leq t_{\max}$ and get the converged Green's function. 2. Extend the time domain by an interval of width $\Delta t$ and prepare the initial guess in the extended domain. 3. Calculate the self-energy in each block. 4. Solve the Dyson equation in each block. 5. Fuse the Green's functions in the four blocks. 6. Perform an affine transformation.

all the QTTs share the same number of bits for the real-time axes, the contraction of two QTTs and the implementation of the linear equation solver are straightforward. We also introduce a cutoff time $t_{\max} + \Delta t$, where $\Delta t$ is relatively small compared to $t_{\max}$. All QTTs are set to zero beyond the cutoff time. In Appendix B, we explain how to set the cutoff time.

To perform the above projections, note that for the retarded and lesser components the different possible values for the most significant bits $(t_1, t_1')$ correspond to the four blocks as follows,

- $(t_1, t_1') = (0, 0)$: past time domain $((1, 1)$-component)

- $(t_1, t_1') = (0, 1)$: off-diagonal time domain $((1, 2)$-component)

- $(t_1, t_1') = (1, 0)$: off-diagonal time domain $((2, 1)$-component)

- $(t_1, t_1') = (1, 1)$: new diagonal time domain $((2, 2)$-component)

On the other hand, for the left-mixing Green's function, which has only one real-time index, the two possible values for the most significant bit $t_1$ correspond to the two blocks:

- $t_1 = 0$: past time domain (component-1)

- $t_1 = 1$: new time domain (component-2)

The above structures allow us to project a QTT on a particular block simply by projecting just the core tensors for the most significant bits.

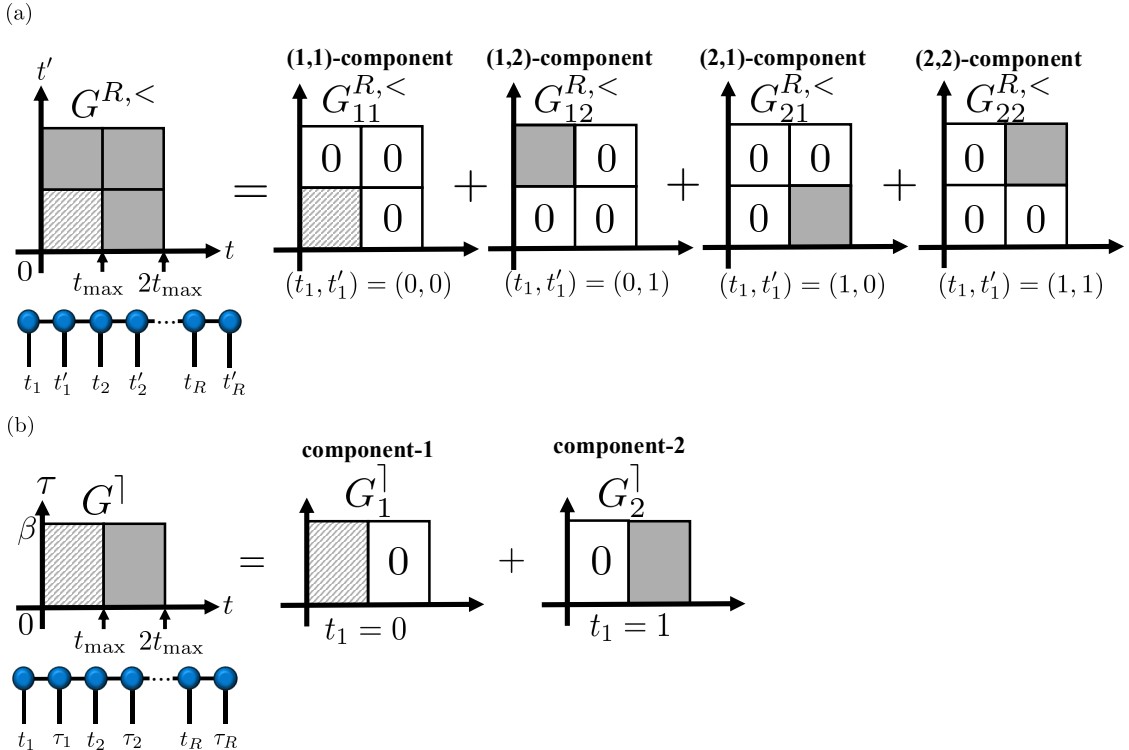

Figure 3: Schematic illustration of how the Green's functions are divided into blocks: (a) For retarded and lesser Green's functions, the time domain is partitioned into four blocks corresponding to $(t_1, t_1') = (0,0), (0,1), (1,0), (1,1)$. (b) For the left-mixing Green's function, the single real-time index is divided into two blocks corresponding to $t_1 = 0$ and $t_1 = 1$.

## 3.2 Procedure for the divide-and-conquer algorithm

Consider Fig. 2. In Step 1, we converge the Green's function $G$ up to $t_{\max}$ via self-consistent iterations using global updates, where we do not use the block structure for the real-time axes.

In Step 2, we initialize the Green's function within the past time domain with the converged Green's function up to $t_{\max}$, while within the extended time domain with the some initial guess such as the non-interacting Green's function. Appendix B details how the initial guess is prepared. Note that the initial guesses for the extended time domain have nonzero values only up to $t_{\max} + \Delta t$.

In Steps 3 and 4, we compute the self-energy and the Green's function on the extended time domain self-consistently by updating them in turn until convergence. This is done in a block-wise manner: we compute only the three QTTs for the three blocks of the extended time domain (for the retarded and lesser Green's functions; two blocks for the left-mixing component). Thanks to the cutoff time $t_{\max} + \Delta t$ applied in the initial guesses, the converged solutions have nonzero values only for $t_{\max} < t, t' \leq t_{\max} + \Delta t$. See Appendix C for details on how to solve the Dyson equation using the linear equation solver in a block-wise manner. Crucially, during this self-consistent iteration, we do not recalculate or alter the objects in the past time domain (shaded region in Steps 3 and 4), taking advantage of the causality of the Green's functions.

After the self-consistent iteration, in Step 5, we add the QTTs for all the blocks to obtain a single QTT, $G^{(\text{new})}$, covering the full time domain from 0 to $t_{\max} + \Delta t$. The result is then truncated by an SVD.

In Step 6, we shrink the real-time domain of the QTTs obtained in Step 5 to fit the whole time domain $0 \leq t, t' \leq t_{\max} + \Delta t$ into the first block. This can be done by an affine transformation with a scaling factor $(t_{\max} + \Delta t)/t_{\max}$. This transformation also rescales the time step $h_t$ by a factor of $(t_{\max} + \Delta t)/t_{\max}$, i.e., $h_t \to h_t \cdot (t_{\max} + \Delta t)/t_{\max}$. The affine transformation can be done by applying an MPO of small bond dimension to the QTTs for $G^{(\text{new})}$ [56], as implemented in the Quantics.jl package [64, 65].

Finally, we return to Step 2 by replacing $t_{\max}$ by $t_{\max} + \Delta t$ and repeat the procedure until $t_{\max}$ reaches the desired value.

In the following, we refer to this series of approaches as "block time stepping." Block time stepping partitions the time domain into blocks, and updates the Green's functions only within the newly added blocks, while keeping the past blocks fixed.

# 4  Hubbard model and nonequilibrium DMFT

In this paper, we consider the half-filled Hubbard model on the Bethe lattice with infinite coordination number ($z \to \infty$). The Hamiltonian is

$$\hat{H}(t) = -J \sum_{\langle i,j \rangle, \sigma} \hat{c}_{i\sigma}^\dagger \hat{c}_{j\sigma} + U(t) \sum_i \left( \hat{n}_{i\uparrow} - \frac{1}{2} \right) \left( \hat{n}_{i\downarrow} - \frac{1}{2} \right), \tag{20}$$

where $\hat{c}_{i\sigma}^\dagger$ ($\hat{c}_{i\sigma}$) is the creation (annihilation) operator for an electron at site $i$ with spin $\sigma$, and $\hat{n}_{i\sigma} = \hat{c}_{i\sigma}^\dagger \hat{c}_{i\sigma}$. $-J = -J^*/\sqrt{z}$ is the hopping integral between nearest-neighbor sites with the renormalized hopping amplitude $J^*$. $U(t)$ is the time-dependent on-site Hubbard interaction.

In the following, we set $\hbar = 1$ and take $J^*$ and $1/J^*$ as units of energy and time, respectively. Thus, the density of states of the Bethe lattice is $\rho(\epsilon) = \sqrt{4 - \epsilon^2}/2\pi$ with bandwidth $W = 4$. Our time unit corresponds to the electron hopping timescale, which is a few femtoseconds in typical materials.

In this work, we study the quench dynamics of antiferromagnetic (AFM) states with nonequilibrium DMFT formulated on the L-shaped contour $\mathcal{C}$ (Fig. 1(a)) [35, 66–68]. In DMFT, the lattice self-energy is assumed to be local, and the local self-energy and the local lattice Green's function are identified with those of the effective impurity model, i.e., $\Sigma_{ij}^{\text{latt}} = \delta_{ij} \Sigma^{\text{imp}}$ and $G_{ii}^{\text{latt}} = G^{\text{imp}}$, which becomes exact in the limit $z \to \infty$. We define the local Green's function as

$$G_\sigma(z, z') = -i \langle T_\mathcal{C} \hat{c}_\sigma(z) \hat{c}_\sigma^\dagger(z') \rangle, \tag{21}$$

where $T_\mathcal{C}$ is the time-ordering operator on the contour $\mathcal{C}$. This Green's function satisfies the nonequilibrium Dyson equation,

$$G_\sigma(z, z') = \mathcal{G}_\sigma^{\text{MF}}(z, z') + \left[ \mathcal{G}_\sigma^{\text{MF}} * \Sigma_\sigma * G_\sigma \right](z, z'), \tag{22}$$

where $*$ denotes the convolution integral on the contour $\mathcal{C}$ and $\Sigma_\sigma$ is the electron self-energy at the impurity site. The Weiss Green's function $\mathcal{G}_\sigma^{\text{MF}}$ for the impurity problem satisfies

$$\left[ i\partial_z - h_\sigma^{\text{MF}}(z) \right] \mathcal{G}_\sigma^{\text{MF}}(z, z') - \left[ \Delta_\sigma * \mathcal{G}_\sigma^{\text{MF}} \right](z, z') = \delta_\mathcal{C}(z, z'), \tag{23}$$

where $\delta_\mathcal{C}$ is the Dirac delta function on the contour $\mathcal{C}$ and $\Delta_\sigma$ is the hybridization function. The mean-field term $h_\sigma^{\text{MF}}$ is defined as $h_\sigma^{\text{MF}}(z) = U(z)(n_{\bar{\sigma}}(z) - 1/2)$ with $n_{\bar{\sigma}}$ the number of electrons of spin $\bar{\sigma}$. ($\bar{\sigma}$ denotes the spin opposite to $\sigma$, i.e., $\bar{\sigma} = \downarrow$ if $\sigma = \uparrow$.) The mean-field self-energy can be written as $\Sigma_\sigma^{\text{MF}}(z, z') = h_\sigma^{\text{MF}}(z)\delta_\mathcal{C}(z, z')$. On the Bethe lattice, the hybridization function can be directly calculated from the impurity Green's function as $\Delta_\sigma(z, z') = J^* G_{\bar{\sigma}}(z, z')J^*$. In

this study, we focus on the weak-coupling regime. Therefore, we approximate the impurity self-energy with second-order perturbation theory [35,68],

$$\Sigma_\sigma(z,z') = U(z)G_{\bar{\sigma}}(z,z')G_{\bar{\sigma}}(z',z)G_\sigma(z,z')U(z'). \tag{24}$$

Note that this is a conserving approximation. Therefore, the total energy is conserved after the system is excited. We can use the conservation of total energy as a criterion for the convergence of the self-consistent loop [35].

We introduce the inverse of the Green's function, $\left(g_\sigma^{\mathrm{MF}}\right)^{-1}(z,z') = \left[i\partial_z - h_\sigma^{\mathrm{MF}}(z)\right]\delta_{\mathcal{C}}(z,z')$, and solve the following Dyson equation,

$$G_\sigma(z,z') = g_\sigma^{\mathrm{MF}}(z,z') + \left[g_\sigma^{\mathrm{MF}} * (\Delta_\sigma + \Sigma_\sigma) * G_\sigma\right](z,z'), \tag{25}$$

which combines Eq. (22) and Eq. (23) into a single expression. The Green's function $g_\sigma^{\mathrm{MF}}$ explicitly reads

$$g_\sigma^{\mathrm{MF}}(z,z') = -i\left[\theta_{\mathcal{C}}(z,z') - f(h_\sigma^{\mathrm{MF}}(0_-))\right]e^{-i\int_{z'}^{z} d\bar{z} h_\sigma^{\mathrm{MF}}(\bar{z})}, \tag{26}$$

where $\theta_{\mathcal{C}}$ is the Heaviside step function on the contour $\mathcal{C}$ and $f(\epsilon) = 1/(\exp(\beta\epsilon)+1)$ is the Fermi-Dirac distribution function with inverse temperature $\beta$. $h_\sigma^{\mathrm{MF}}(0_-)$ corresponds to the equilibrium value of the mean-field term. In the nonequilibrium case, the mean-field term $h_\sigma^{MF}$ is time-dependent. Therefore, $g_\sigma^{\mathrm{MF}}$ typically cannot be expressed in a simple QTT form. In our case, where the number of time steps is very large, it is impractical to obtain the TT representation via the SVD using all $(z,z')$ elements of $g_\sigma^{\mathrm{MF}}$. Instead, we use tensor cross interpolation (TCI) [46,48,55,69], which constructs a TT approximation by sampling a subset of the elements of the original tensor.

## 5 Results

In this study, we assess the convergence of the self-consistent loop by monitoring the following convergence errors:

- Imaginary time:
  $$\epsilon_{\mathrm{conv},\sigma} = \frac{\|G_\sigma^{(\mathrm{new})M} - G_\sigma^{(\mathrm{old})M}\|_{\mathrm{F}}}{\|G_\sigma^{(\mathrm{old})M}\|_{\mathrm{F}}},$$

- Real time:
  $$\epsilon_{\mathrm{conv},\sigma} = \max_{\alpha=R,],<}\left\{\frac{\|G_\sigma^{(\mathrm{new})\alpha} - G_\sigma^{(\mathrm{old})\alpha}\|_{\mathrm{F}}}{\|G_\sigma^{(\mathrm{old})\alpha}\|_{\mathrm{F}}}\right\}.$$

$\|*\|_{\mathrm{F}}$ denotes the Frobenius norm. We define the total convergence error as $\epsilon_{\mathrm{conv}} = \max\left\{\epsilon_{\mathrm{conv},\uparrow}, \epsilon_{\mathrm{conv},\downarrow}\right\}$. In numerical calculations, these errors are influenced by the cutoff parameter $\epsilon_{\mathrm{cutoff}}$ and the maximum bond dimension $D$. Note that $\epsilon_{\mathrm{cutoff}}$ is defined as $\|A - \tilde{A}\|_{\mathrm{F}}^2/\|A\|_{\mathrm{F}}^2 < \epsilon_{\mathrm{cutoff}}$ [48,63], where $A$ and $\tilde{A}$ are the exact and truncated tensors, respectively.

For the self-consistent calculation of the Matsubara Green's function, we set $\epsilon_{\mathrm{cutoff}} = 10^{-20}$ and the imaginary time step $h_\tau \approx 5.7 \times 10^{-13}$. In the present study, we achieve convergence at an error level of $\epsilon_{\mathrm{conv}} \sim 10^{-9}$. At this high accuracy, the maximum bond dimension of the Matsubara Green's function is about 10.

In this section, we compare the results of our method with those obtained by the conventional approach implemented with the NESSi library [6], which solves the integro-differential equations for the Green's functions (the Kadanoff-Baym equations) in a causality-preserving manner. In the calculation with NESSi, we use $h_t = 0.02$ and $h_\tau = 0.02$ for the real and imaginary time steps, respectively. These values are sufficiently small to ensure the convergence of the results as NESSi uses integration with high-order quadrature rules [6].

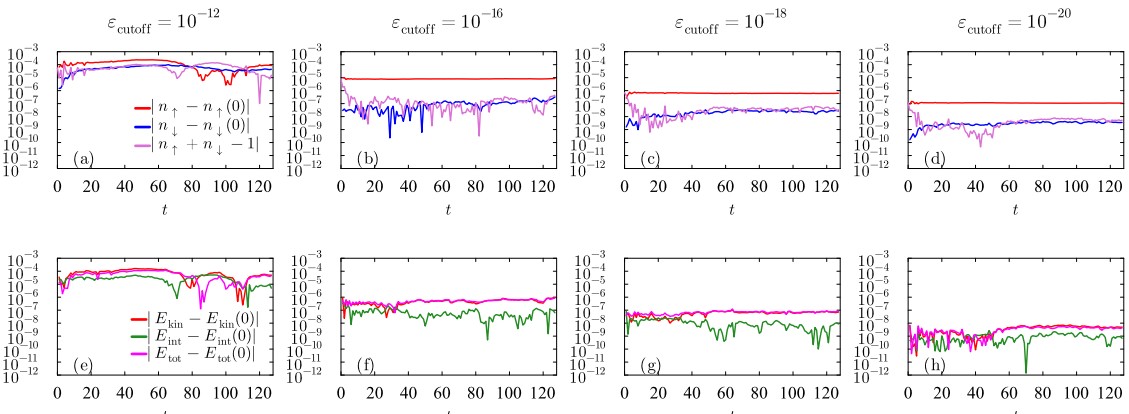

Figure 4: Particle number and energy conservation in the equilibrium AFM state. (a-d) The absolute error from the initial state $t = 0$ for $n_\uparrow$, $n_\downarrow$, and $n_\uparrow + n_\downarrow - 1$. (e-h) The absolute error from the initial state $t = 0$ of the electron kinetic, interaction, and total energies $E_{\mathrm{kin}}$, $E_{\mathrm{int}}$, and $E_{\mathrm{tot}}$, respectively. (a) and (e) are for $\epsilon_{\mathrm{cutoff}} = 10^{-12}$, $\epsilon_{\mathrm{conv}} \sim 10^{-4}$. (b) and (f) are for $\epsilon_{\mathrm{cutoff}} = 10^{-16}$, $\epsilon_{\mathrm{conv}} \sim 6 \times 10^{-7}$. (c) and (g) are for $\epsilon_{\mathrm{cutoff}} = 10^{-18}$, $\epsilon_{\mathrm{conv}} \sim 8 \times 10^{-8}$. (d) and (h) are for $\epsilon_{\mathrm{cutoff}} = 10^{-20}$, $\epsilon_{\mathrm{conv}} \sim 5 \times 10^{-9}$.

## 5.1 Equilibrium

First, we test the QTT-NEGF method by calculating real-time Green's functions in equilibrium. We use global updates with the linear equation solver to obtain the Green's functions for $t \leq 128$. We set the real time step to $h_t \approx 2.8 \times 10^{-14}$. In equilibrium, physical observables should remain unchanged from the initial state ($t = 0$). Therefore, we can check the accuracy of our method by estimating the deviation of observables from their values at $t = 0$. Because the number of electrons $n_\sigma$ should not change in principle, we can fix $n_\sigma$ to the value calculated using the Matsubara Green's function. Therefore, we do not need to update $g_\sigma^{\mathrm{MF}}$ (Eq. (26)). In this subsection, we set $\beta = 20$ and $U = 2$. With these parameters, the system is in the AFM state [35]. In the following calculation, we fix the maximum bond dimension to $D = 80$. Starting from Green's functions that were converged to $\epsilon_{\mathrm{conv},\sigma} = 10^{-4}$ with $\epsilon_{\mathrm{cutoff}} = 10^{-12}$, we successively perform 400 iterations with $\epsilon_{\mathrm{cutoff}} = 10^{-16}$, followed by 400 iterations with $\epsilon_{\mathrm{cutoff}} = 10^{-18}$, and then 400 iterations with $\epsilon_{\mathrm{cutoff}} = 10^{-20}$. As a result, $\epsilon_{\mathrm{conv},\sigma}$ is eventually reduced to approximately $5 \times 10^{-9}$.

Figure 4 shows how the observable errors are improved when the cutoff parameter $\epsilon_{\mathrm{cutoff}}$ and the convergence parameter $\epsilon_{\mathrm{conv},\sigma}$ are varied. We calculate the number of electrons $n_\sigma(t) = -iG_\sigma^<(t,t)$, the AFM order parameter $m(t) = n_\uparrow(t) - n_\downarrow(t)$, the kinetic energy $E_{\mathrm{kin}}(t) = \sum_\sigma -i[\Delta_\sigma * G_\sigma]^<(t,t)$, and the interaction energy $E_{\mathrm{int}}(t) = \sum_\sigma (-i/2)[(\Sigma_\sigma^{\mathrm{MF}} + \Sigma_\sigma) * G_\sigma]^<(t,t)$. The total energy is defined as $E_{\mathrm{tot}}(t) = E_{\mathrm{kin}}(t) + E_{\mathrm{int}}(t)$. The parameter settings are described in the caption of Fig. 4. As $\epsilon_{\mathrm{cutoff}}$ and $\epsilon_{\mathrm{conv},\sigma}$ are decreased, the errors in all observables decrease. This demonstrates that the accuracy of physical quantities can be systematically controlled by adjusting the cutoff parameter $\epsilon_{\mathrm{cutoff}}$ (and the convergence error $\epsilon_{\mathrm{conv},\sigma}$). Figure 5 compares the physical quantities calculated with the QTT-NEGF method to those obtained using the conventional approach implemented with NESSi [6] (gray dashed lines in Fig. 5(a) and 5(b)) for $\epsilon_{\mathrm{cutoff}} = 10^{-20}$ and $\epsilon_{\mathrm{conv}} \sim 5 \times 10^{-9}$. We also examine the error relative to the data calculated with NESSi in Fig. 5(c) and 5(d). The results obtained by our method are in good agreement with the reference data.

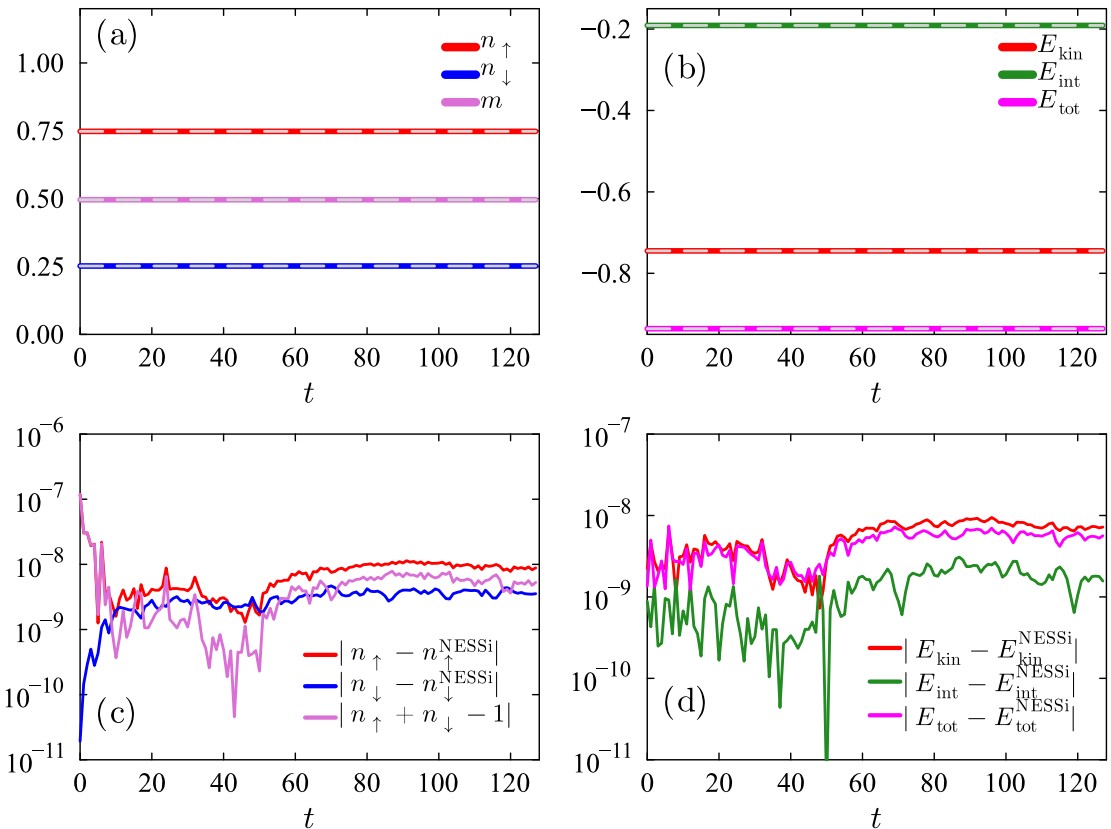

Figure 5: Physical quantities in the equilibrium AFM state for the QTT-NEGF and conventional methods. (a) The number of electrons with up and down spins, $n_\uparrow$ and $n_\downarrow$, and the order parameter $m = n_\uparrow - n_\downarrow$. (b) The electron kinetic, interaction, and total energies, $E_{\mathrm{kin}}$, $E_{\mathrm{int}}$, and $E_{\mathrm{tot}}$, respectively. The gray dashed lines are the reference data calculated by the conventional method using the NESSi [6]. (c) The absolute error of the number of electrons between the results of QTT-NEGF and NESSi. We also plot the absolute error of $n_\uparrow + n_\downarrow - 1$. (d) The absolute error of energy between the results of QTT-NEGF and NESSi.

## 5.2 Quench dynamics

We consider the quench dynamics, where the on-site Hubbard interaction is suddenly changed from $U = 2$ to $1.5$ at $t = 0$. In the AFM state, the relaxation time to the new equilibrium state is determined by the excitation condition—in our case, the final value of $U$ [35, 66–68]. Here, we focus on the relatively slow relaxation dynamics of the order parameter. The application of our method to the quench dynamics of paramagnetic (PM) states [68, 70, 71] is discussed in Appendix E.

First, we solve the nonequilibrium Dyson equation using the global update scheme to obtain the Green's functions up to $t_{\mathrm{max}} = 128$ (see Sec. 5.2.1). Next, starting from this converged Green's function for $t \leq 128$, we extend the time domain to $t_{\mathrm{max}} \approx 300$ using block time stepping based on the divide-and-conquer algorithm (see Sec. 5.2.2). Below, we describe each step in detail.

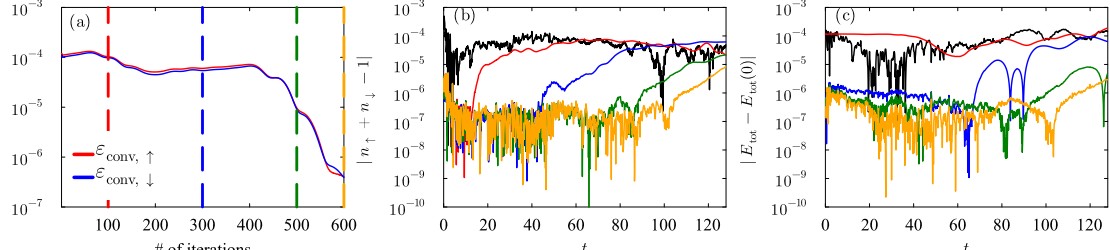

Figure 6: Convergence properties of global updates for a quench from $U = 2$ to $U = 1.5$ in the AFM state of the Hubbard model. (a) Change in the convergence error $\epsilon_{\mathrm{conv},\sigma}$ for real-time Green's functions as a function of the number of iterations. (b) Absolute error from the initial state $t = 0$ of $n_\uparrow + n_\downarrow - 1$ as a function of the number of iterations. (c) Absolute error from the initial state $t = 0$ of $E_{\mathrm{tot}}$ as a function of the number of iterations. Each colored line in (b) and (c) corresponds to the same color dashed line in panel (a), i.e. to a given number of iterations.

### 5.2.1 Results of the global-update method

First, we simulate the quench dynamics using the global update with the linear equation solver [48]. As in the equilibrium calculations, we set the real time step to $h_t \approx 2.8 \times 10^{-14}$ and fix the maximum bond dimension to $D = 80$. For reference, when compressing Green's function data (up to $t_{\mathrm{max}} = 320$) obtained by the conventional method [6] using the SVD, the maximum bond dimension is about 70 for $\epsilon_{\mathrm{cutoff}} = 10^{-14}$ and about 90 for $\epsilon_{\mathrm{cutoff}} = 10^{-16}$. Therefore, fixing $D = 80$ is reasonable for capturing the structure of the Green's function.

Figure 6(a) shows how $\epsilon_{\mathrm{conv},\sigma}$ changes with the number of iterations for $\epsilon_{\mathrm{cutoff}} = 10^{-16}$. Here, we use the linear mixing method with a mixing ratio $\alpha = 0.5$ for updating the Green's functions. We start from an initial guess of the Green's function precomputed with $\epsilon_{\mathrm{conv},\sigma} \sim 10^{-4}$ (black lines in Fig. 6(b),(c)). This initial guess is already close to the reference data calculated by the conventional method, although it slightly deviates from it near $t_{\mathrm{max}}$ (see Appendix D). Until the number of iterations reaches about 400, the convergence errors do not decrease significantly (plateau region), but after that, the errors begin to decrease. Interestingly, in the plateau region, the absolute errors of $n_\uparrow + n_\downarrow - 1$ and $E_{\mathrm{tot}} - E_{\mathrm{tot}}(0)$ decrease while $\epsilon_{\mathrm{conv},\sigma}$ is approximately constant. This improvement in accuracy is observed starting from the short-time domain. This is reasonable because the accuracy in the long-time domain depends on the accuracy in the past time domain, which enters through the convolution integral. When the number of iterations reaches 600, $\epsilon_{\mathrm{conv},\sigma}$ decreases to $\sim 4.0 \times 10^{-7}$ (orange lines in Fig. 6(b),(c)) and the results reach a very good agreement with the reference data (see Appendix D). These results indicate that even when one can accurately converge the Green's function with global updates, it takes an excessive number of iterations to do so, irrespective of the fact that the initial guess is close to the exact solution.

### 5.2.2 Results of the block time stepping

Next, we examine the accuracy and efficiency of block time stepping based on the divide-and-conquer algorithm. We set the initial $t_{\mathrm{max}} = 128$ and use the precomputed Green's function from Sec. 5.2.1 as the starting point, corresponding to the $(1, 1)$-component in Fig. 3. In each block time stepping, the time domain is extended by a factor of $(t_{\mathrm{max}} + \Delta t)/t_{\mathrm{max}} = 1.1$. As the initial guess for the Green's function in the extended time domain, we use the non-interacting Green's function calculated with the density of states of the Bethe lattice, $G = \int_{-\infty}^{\infty} d\epsilon\, \rho(\epsilon) G_\epsilon^0$, where $\epsilon$ is the eigenenergy of the tight-binding model (i.e., the $U = 0$ case in Eq. (20)).

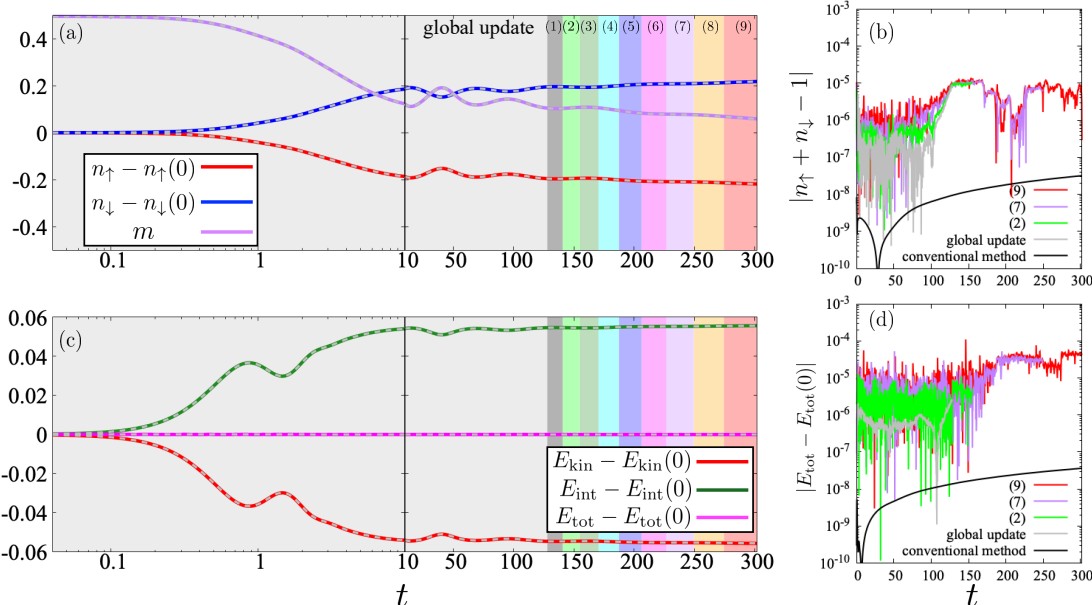

Figure 7: Time evolution of physical observables obtained with the divide-and-conquer algorithm for a quench from $U = 2$ to $U = 1.5$ in the AFM state of the Hubbard model. (a) Time evolution of $n_\uparrow(t) - n_\uparrow(0)$, $n_\downarrow(t) - n_\downarrow(0)$, and $m$. (b) The absolute error of $n_\uparrow + n_\downarrow - 1$. (c) Time evolution of $E_{\text{kin}}(t) - E_{\text{kin}}(0)$, $E_{\text{int}}(t) - E_{\text{int}}(0)$, and $E_{\text{tot}}(t) - E_{\text{tot}}(0)$. (d) The absolute error of $E_{\text{tot}}$ relative to the initial state $t = 0$. Colored regions indicate the time intervals added in each block time stepping. The gray dashed lines in (a) and (c) are the reference data calculated by the conventional method using the NESSi library [6].

Figures 7(a) and 7(c) show the time evolution of physical observables obtained with the block time stepping based on a divide-and-conquer algorithm. The colored regions indicate the time intervals added in each block time stepping. After 9 extension steps, $t_{\text{max}}$ reaches $\approx 300$, which is $1.1^9$ times larger than the initial value $t_{\text{max}} = 128$. The results up to $t_{\text{max}} \approx 300$ are in good agreement with the reference data for all physical observables (gray dashed lines).

Figures 7(b) and 7(d) show the absolute errors of $n_\uparrow + n_\downarrow - 1$ and $E_{\text{tot}} - E_{\text{tot}}(0)$, respectively. The conservation of the total number of particles and the total energy is maintained with an accuracy better than $10^{-4}$ (red lines in Figs. 7(b) and 7(d)). Although this accuracy is lower than that achieved by the conventional method [6] (black lines), it is sufficient for the observables studied here.

Note that the accuracy of the total number of particles $n_\uparrow + n_\downarrow$ and the total energy $E_{\text{tot}}$ in the past time domain ($t \leq 128$) deteriorates as the number of extensions of the time domain increases (compare the gray and red lines in Figs. 7(b) and 7(d)). This is because after we perform the affine transformation, we truncate the QTT with a bond dimension $D = 80$ to suppress the bond dimension growth. At this level of accuracy, we can extend the time domain up to $t_{\text{max}} \approx 300$ as shown in Fig. 7. When truncating to $D = 150$ after the affine transformation, we confirmed that the accuracy in the past time domain is maintained at the original level.

Figure 8 shows how the convergence error $\epsilon_{\text{conv}}$ decreases with the number of iterations in the self-consistent loop. Each colored line represents the error for the calculation performed in the corresponding colored region shown in Fig. 7. Only the extended time domain is repeatedly updated, while the past data in the $(1, 1)$ block is kept fixed. First, we update the Green's function with the cutoff $\epsilon_{\text{cutoff}} = 10^{-12}$ and the maximum bond dimension $D = 50$ until $\epsilon_{\text{conv},\sigma}$ reaches $5.0 \times 10^{-5}$. Afterwards, we change the cutoff to $\epsilon_{\text{cutoff}} = 10^{-16}$ and iterate the self-

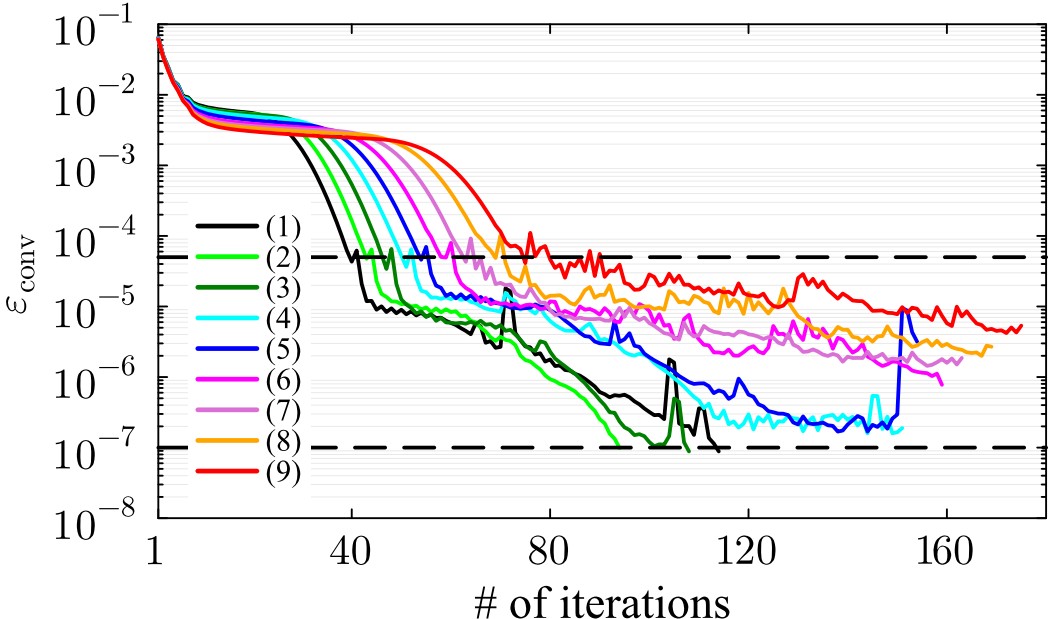

Figure 8: Change in the convergence error $\epsilon_{\text{conv}}$ for the real-time Green's functions as a function of the number of iterations. The colored lines correspond to the colored regions shown in Fig. 7. After $\epsilon_{\text{conv}}$ reaches $5.0 \times 10^{-5}$, we change the cutoff from $10^{-12}$ to $10^{-16}$ and iterate the self-consistent loop 100 times or until $\epsilon_{\text{conv}}$ reaches $1.0 \times 10^{-7}$.

consistent loop 100 times or until $\epsilon_{\text{conv}}$ reaches $1.0 \times 10^{-7}$ (steps 3 and 4 in Fig. 2). Note that $D$ is fixed at 50 only for the calculations in the extended time domain, and the bond dimension of the Green's function in the past time domain is maintained at $D = 80$. Reducing the maximum bond dimension in the new time domain allows $\epsilon_{\text{conv}}$ to converge with reduced computational time. As the time domain is extended, more iterations are required for $\epsilon_{\text{conv}}$ to reach the prescribed convergence criterion. This is because in our calculation, the size of the extended time domain becomes larger by a factor of $(t_{\max} + \Delta t)/t_{\max}$ as the number of extensions increases.

Figure 9 compares the data size of the Green's functions between the conventional method and our QTT-NEGF approach. Note that these values do not represent the actual runtime memory usage during the calculations. For example, in solving the Dyson equation and computing diagrams, the runtime memory usage is typically estimated as $\mathcal{O}(LD^3)$ [48], and in practice it can be about $D$ times larger than the data size, which is $\mathcal{O}(LD^2)$, where $L$ is the length of the MPOs (see Sec. 2).

For the conventional method, the data size is $(t_{\max}/h_t + 1)^2 \times 16/(1024)^3$ gigabytes (GB) for the retarded and lesser components, and $(t_{\max}/h_t + 1) \times (\beta/h_\tau + 1) \times 16/(1024)^3$ GB for the left-mixing component (including the element at $t = 0$). Here, we assume $h_t = 0.02$ and $h_\tau = 0.02$. In our QTT implementation, the data size is calculated as 16 bytes times the total number of elements in all core tensors. While the conventional method fixes the time step $h_t = 0.02$ and increases the number of time steps $N_t$, our QTT calculation fixes the number of time steps at $N_t = 2^{52} - 1$ and increases the time step $h_t$ by a factor of $(t_{\max} + \Delta t)/t_{\max}$ with each extension of the time domain. Note that due to the huge number of time steps that we

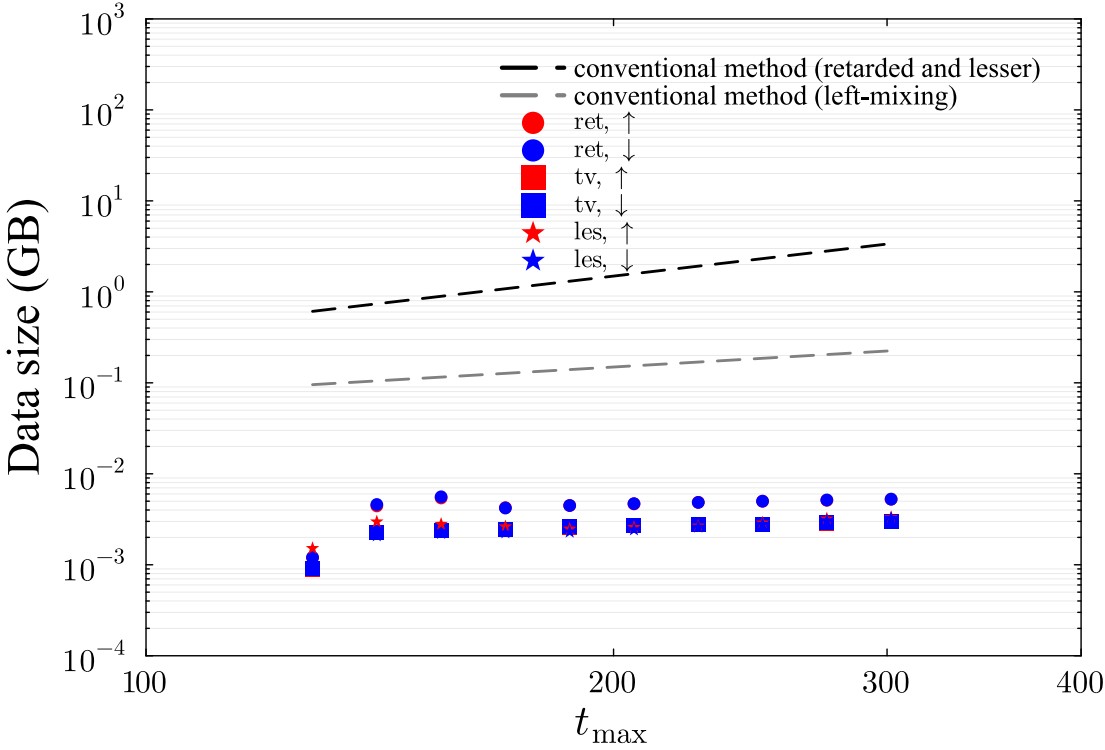

Figure 9: Data size of the Green's functions. Black dashed line shows the data size of retarded (ret) and lesser (les) Green's functions, and gray dashed line shows the data size of left-mixing (tv) Green's function, in the conventional method with $h_t = 0.02$ and $h_\tau = 0.02$.

use, our $h_t$ always remains small enough to have negligible discretization errors, even when integrating with the trapezoidal rule.

In the conventional approach, the data size of the lesser (retarded) and left-mixing components increases proportionally to $t_{\max}^2$ and $t_{\max}$, respectively (see black and gray dashed lines in Fig. 9). For example, the data size of the lesser Green's function exceeds 3 GB at $t_{\max} \sim 300$. In contrast, with our QTT method, the data size of the Green's functions remains at most 0.0053 GB even at $t_{\max} \sim 300$, and the increase in data size with $t_{\max}$ is significantly suppressed compared to the conventional approach. These results indicate that the causality-based method enables a stable extension of the time domain without a significant increase in the data size.

Finally, in Appendix F, we present simulation results for further extension of the time domain beyond $t_{\max} \approx 300$. Using the same range in y axes as in Figs. 7(a) and 7(c), we confirm that the total energy remains well conserved and the results agree with the reference data up to $t_{\max} \approx 534$. However, as the time domain is extended, the accuracy of the total particle number and total energy gradually deteriorates.

# 6 Conclusion

In this work, we proposed a causality-based divide-and-conquer algorithm for QTT-NEGF calculations. We combined this algorithm with nonequilibrium DMFT and applied it to the simulation of quench dynamics in the AFM state of the Hubbard model. First, we studied the convergence behavior of the Green's functions using the global update scheme (without exploiting the causality) and found that, even when starting from a good initial guess close to the true solution, many iterations are required to achieve convergence. During these iterations, the accuracy of the Green's function improves first at short and only then at long time scales. Next, we applied the divide-and-conquer algorithm to extend the time domain from $t_{\max} = 128$ to $t_{\max} \approx 300$. We demonstrated that the time domain can be gradually extended without causing instabilities or significant slowdowns in the convergence. By recompressing the functions after each extension of the time domain, we can also avoid a significant increase in data size.

The causality-based approach allowed us to apply QTT-NEGF with DMFT for the first time to long-time dynamics of symmetry-broken (ordered) states, confirming its effectiveness in this setting. The systematic extension of the simulated time domain is crucial to accurately capture slow relaxation processes characteristic of symmetry-broken phases [35, 42, 66, 67]. We expect that our causality-based method will be useful for studying long-time relaxation dynamics in various ordered states.

There is still room for improvement of the present divide-and-conquer algorithm. In this study, we extended the time domain by a factor of $(t_{\max} + \Delta t)/t_{\max}$ in each block time stepping. As shown in our results, the number of iterations required to achieve the convergence criterion increased as the time domain was extended. A more efficient approach is to extend the time domain by a fixed small increment $\Delta t$ in each block time stepping, using a masking function (see Appendix B for details). Another point worth improving is the initial guess in the extended time domain. We used the non-interacting Green's function, which is not optimal and requires several hundred iterations to reach convergence, as it is evident from our results. A better initial guess, for example using a suitable extrapolation method, could reduce the number of required iterations. Details of such an improved scheme will be discussed in a future work [72].

# Acknowledgements

KI is grateful to M. Eckstein, Y. Murakami, R. Sakurai, N. Dasari, Y. Nomura, R. Akashi, Y. Michishita, T. Miki, and H. Ishida for fruitful discussions. The QTT-NEGF implementation [48] is written in Julia [73] and is based on the ITensor library [63] and libraries developed by the tensor4all collaboration [46, 64]. The reference data in this paper were calculated with NESSi [6].

**Funding information** KI was supported by JSPS KAKENHI Grant Nos. 23KJ0883 and 25K17307, Japan. HS was supported by JSPS KAKENHI Grant Nos. 22KK0226, 23H03817 as well as JST FOREST Grant No. JPMJFR2232, Japan. This work was supported by JSPS Bilateral Program Number JPJSBP220259901. This work was partly supported by the Austrian Science Fund (FWF) through Grant 10.55776/V1018. MŚ and PW acknowledge support from SNSF Grant No. 200021-196966.

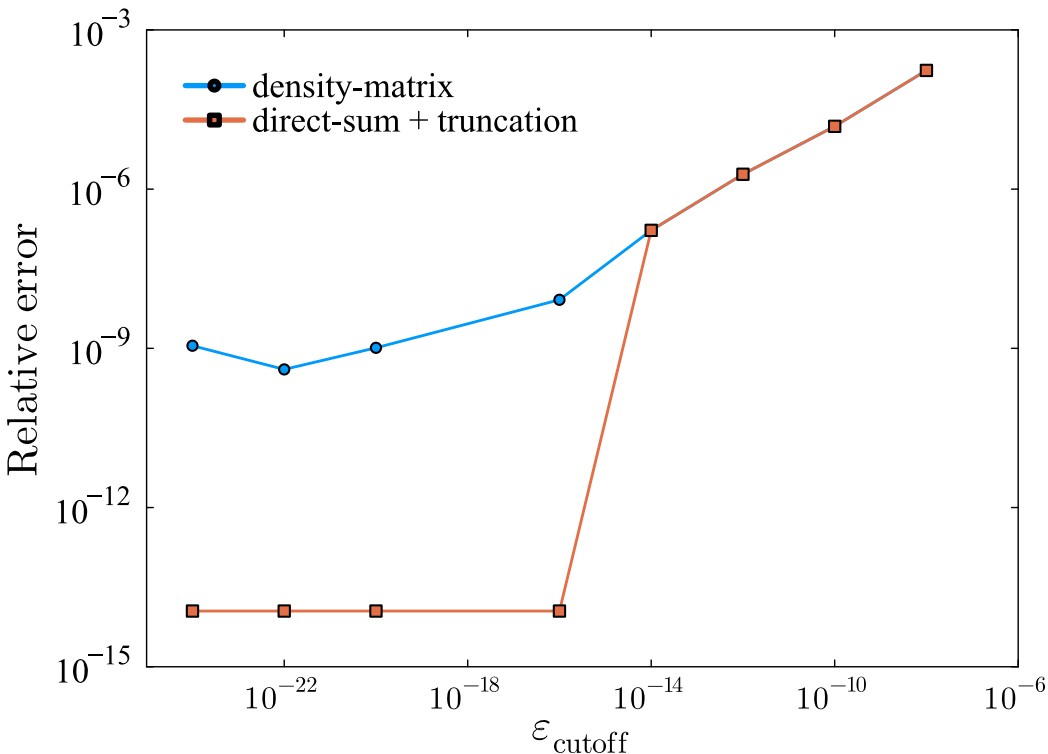

Figure 10: Comparison of the accuracy of the density-matrix and directsum algorithms for the summation of Green's functions.

# A   Comparison of algorithms for the summation of Green's functions

In tensor-network methods, the summation of TTs is often performed using the "density-matrix" algorithm, which is the default algorithm implemented in the ITensor library [63]. In this algorithm, the singular values of the density matrix (i.e., the product of the state vector, TT) are truncated with a cutoff parameter $\epsilon_{\text{cutoff}}$, which may underestimate the actual singular values of the TT. To confirm this, in this Appendix, we compare the "density-matrix" method with the "direct-sum" method. In the "direct-sum" algorithm, the result of the summation can be obtained exactly without the cutoff parameter, but the bond dimension increases. Therefore, an SVD truncation with an appropriate cutoff is necessary and is performed here.

We prepare the non-interacting retarded Green's function $G_0^R$ with eigenenergy $\epsilon = -1.0$ ($D = 3$). Multiplying $G_0^R$ by a scalar does not change its bond dimension, and the result can be obtained exactly. Here, we multiply $G_0^R$ by 10, i.e., $\tilde{G} = 10 G_0^R$ ($D = 3$). Alternatively, $\tilde{G}$ can be obtained by summing $G_0^R$ ten times. In Fig. 10, we show the relative error of the result compared to the exact value $\tilde{G} = 10 G_0^R$ for both algorithms. In the "density-matrix" algorithm, we use $\epsilon_{\text{cutoff}}$ in the summation, while in the "direct-sum" algorithm, we use the same cutoff parameter in the truncation of the result after the summation. We find that for $\epsilon_{\text{cutoff}} \geq 10^{-14}$, the relative error is nearly the same for both algorithms. However, for $\epsilon_{\text{cutoff}} < 10^{-14}$, the error in the "density-matrix" algorithm saturates around $10^{-9}$ and does not decrease further, while the error in the "direct-sum" algorithm continues to decrease down to $10^{-14}$.

The cutoff parameter $\epsilon_{\text{cutoff}} \approx 10^{-14}$ may be sufficiently small for wave function-based

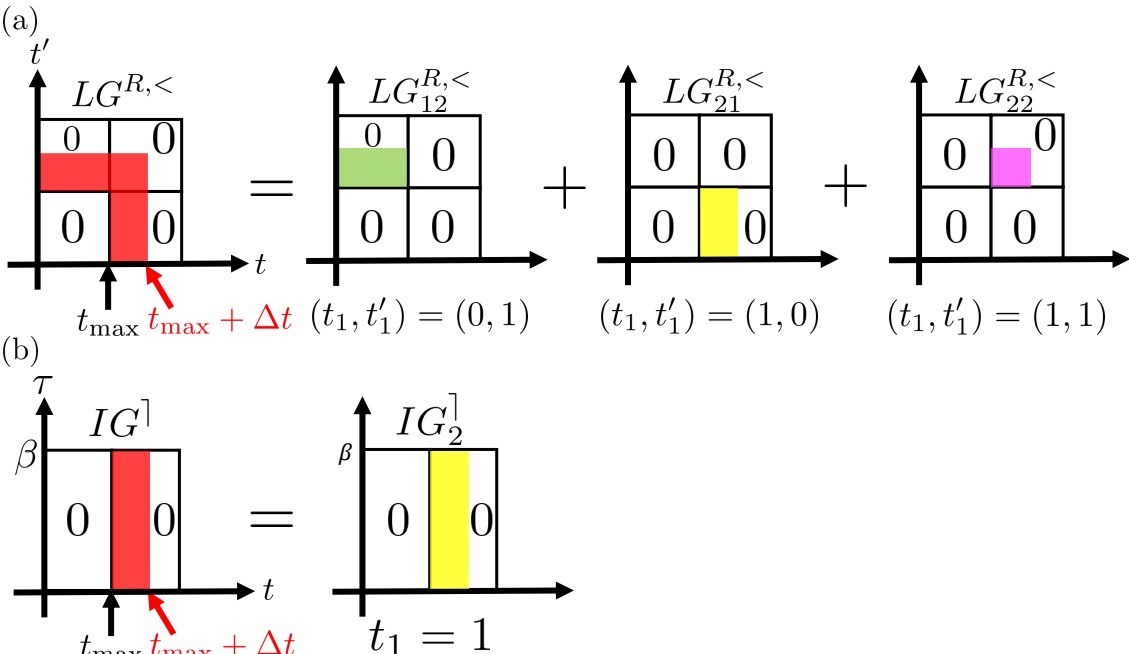

Figure 11: Schematic illustration of how to prepare the initial guess for the Green's function in the extended time domain.

methods such as the DMRG. However, in QTT-NEGF calculations, we need to use a much smaller cutoff parameter to achieve the required accuracy, as shown in the main text. Therefore, we use the "direct-sum" algorithm here. The TT sums are in any case less demanding computationally than convolutions or element-wise multiplications.

## B Preparation of the initial guess for the Green's function in the extended time domain

In this Appendix, we describe how to prepare the initial guess for the Green's function in the extended time domain. By adding extra tensors corresponding to coarse scales, we can extend the time domain by a factor of two. To gradually extend the time domain using a small $\Delta t$, we introduce the following masking functions using the Heaviside step function:

$$M^{R,<}(t,t';t_b) = \theta(t_b - t)\theta(t_b - t'), \tag{B.1}$$

$$M^{\rceil}(t;t_b) = \theta(t_b - t). \tag{B.2}$$

Here, $t_b$ is the boundary time. $M^{R,<}$ and $M^{\rceil}$ can be exactly represented in the QTT format with a maximum bond dimension $D = \mathcal{O}(1)$. Using these masking functions, we define the L- and I-shaped masking functions with width $\Delta t$ as

$$L(t,t') = M^{R,<}(t,t';t_{\max} + \Delta t) - M^{R,<}(t,t';t_{\max}), \tag{B.3}$$

$$I(t) = M^{\rceil}(t;t_{\max} + \Delta t) - M^{\rceil}(t;t_{\max}). \tag{B.4}$$

By element-wise multiplying $L$ and $I$ with a suitable Green's function, e.g., the non-interacting one, we obtain an initial guess that is non-zero solely within the extended time domain (see Fig. 11).

## C Solving the linear equations in each block

In our divide-and-conquer approach, we solve the linear equations separately in each block. The linear operator $A^{\mathrm{rt}}$ and the constant terms $b^{R,\rceil,<}$ are also defined within each block. Below, we summarize the *small* linear equations for the retarded, left-mixing, and lesser components. While we list them for completeness, we do not need to solve the linear equations for the $(1,1)$-component of the retarded and lesser Green's functions, nor for component-1 of the left-mixing Green's function, since these are the already converged past data.

**Retarded component**

- $(1,1)$-component: $A^{\mathrm{rt}}_{11} G^R_{11} = b^R_{11}$

- $(2,1)$-component: $A^{\mathrm{rt}}_{22} G^R_{21} = b^R_{21} - A^{\mathrm{rt}}_{21} G^R_{11}$

- $(2,2)$-component: $A^{\mathrm{rt}}_{22} G^R_{22} = b^R_{22}$

The $(1,2)$-component of the linear operator $A^{\mathrm{rt}}_{12}$ is zero because this operator is constructed with only retarded components.

**Left-mixing component**

- component-1: $A^{\mathrm{rt}}_{11} G^\rceil_1 = b^\rceil_1$

- component-2: $A^{\mathrm{rt}}_{22} G^\rceil_2 = b^\rceil_2 - A^{\mathrm{rt}}_{21} G^\rceil_1$

**Lesser component**

- $(1,1)$-component: $A^{\mathrm{rt}}_{11} G^<_{11} = b^<_{11}$

- $(1,2)$-component: $A^{\mathrm{rt}}_{11} G^<_{12} = b^<_{12}$

- $(2,2)$-component: $A^{\mathrm{rt}}_{22} G^<_{22} = b^<_{22} - A^{\mathrm{rt}}_{21} G^<_{12}$

The off-diagonal block component $G^<_{21}$ satisfies $G^<_{21}(t,t') = -(G^<_{12}(t',t))^*$ ($*$ denotes complex conjugation) [3,6]. Therefore, we do not need to calculate the $(2,1)$-component explicitly.

## D Comparison of order parameters when the number of iterations in the global update is changed

Figure 12 shows the comparison of the order parameter $m$ for different number of global iterations on the contour of length $t_{\max} = 128$, which we later use as the starting point for the divide-and-conquer iterations. Each colored line corresponds to the dashed line of the same color in Fig. 6(a), and hence indicates the result after a given number of iterations. The initial guess (black line), precomputed up to $\epsilon_{\mathrm{conv}} \sim 10^{-4}$ by global iterations, is already close to the reference data (gray line), which is calculated with the conventional method. However, as shown in the insets, this initial guess still deviates slightly from the reference, especially for $t > 80$. After additional 600 global iterations, however, the Green's function is very well converged (orange line) and the order parameter is in good agreement with the NESSi data.

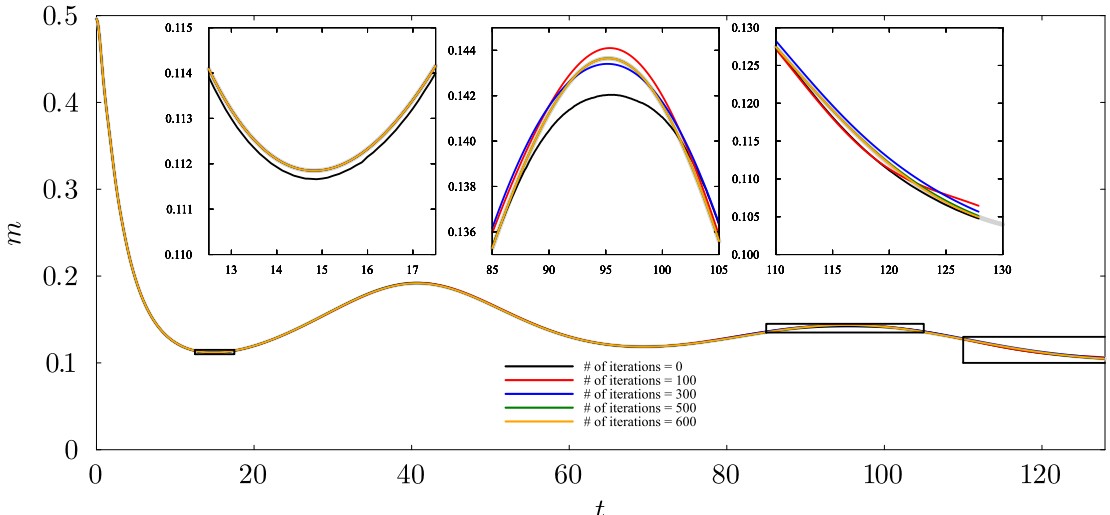

Figure 12: Time evolution of the order parameter $m$ when the on-site interaction is quenched from $U = 2$ to $U = 1.5$. The results are plotted for different numbers of global iterations. The insets show consecutive zoomed-in regions of the plot. The gray dashed lines are the reference data calculated with the conventional method implemented using the NESSi library [6].

## E   Quench dynamics in the PM state

In this Appendix, we test our divide-and-conquer algorithm for the quench dynamics of the PM state, where the Hubbard interaction is changed from $U = 2$ to $U = 1.5$ at $t = 0$. We set $\beta = 20$. In the PM case, the mean-field term vanishes. As in the AFM calculation discussed in the main text, we first calculate the Green's functions up to $t_{\max} = 128$ using global updates with the linear equation solver. We then successively extend $t_{\max}$ by a factor of $(t_{\max} + \Delta t)/t_{\max} = 1.1$. After 15 extension steps, $t_{\max}$ reaches $1.1^{15} \times 128 \approx 534$. Figure 13(a) shows the time evolution of the electron energies, which are in good agreement with the reference data (gray dashed lines). The total energy is well conserved. Note that in the reference NESSi calculations, we set $h_t = 0.04$, which is sufficiently small to ensure convergence. In this fast relaxation case (the system is almost relaxed to the new equilibrium state already for $t \leq 10$), our algorithm enables freely extending the time domain also beyond $t_{\max} = 500$. Similarly as in the AFM case, Fig. 13(b) shows that as the time domain is extended, more iterations are required to reduce the convergence error $\epsilon_{\mathrm{conv}}$ to a prescribed convergence criterion, even though the system appears to already be in a steady state.

## F   Longer-time simulation in the AFM state

To extend the time evolution beyond $t_{\max} \approx 300$ in the AFM state, we perform at most 200 iterations in the self-consistent loop for each new block time stepping (aiming to reach $\epsilon_{\mathrm{conv}} \approx 10^{-4}$). The results are shown in Fig. 14. In the conventional method, we simulate the dynamics beyond $t = 500$ using $h_t = 0.04$ (orange dashed lines). As expected, the accuracy of the conventional simulations is worse for $h_t = 0.04$ than for $h_t = 0.02$ (compare the gray and orange lines in Figs. 14(d) and 14(e)), but still accurate enough to serve as a reference. All physical quantities obtained by QTT-NEGF are in good agreement with the latter reference

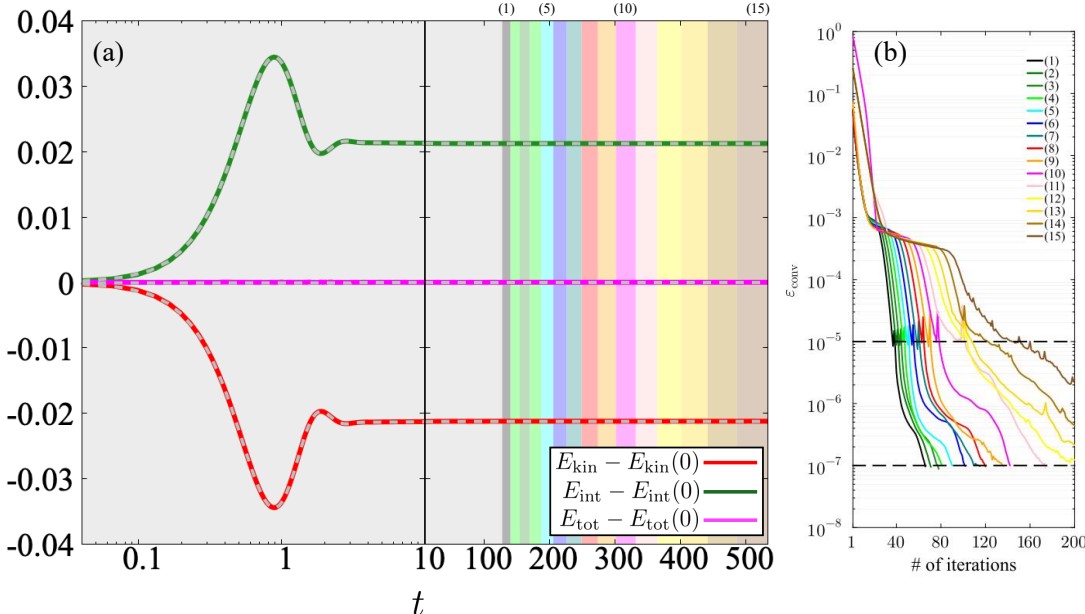

Figure 13: Time evolution of electron energies obtained with the divide-and-conquer algorithm for a quench from $U = 2$ to $U = 1.5$ in the PM state of the Hubbard model. (a) Time evolution of $E_{kin}(t) - E_{kin}(0)$, $E_{int}(t) - E_{int}(0)$, and $E_{tot}(t) - E_{tot}(0)$. Colored regions indicate the increase of the time interval in each block time stepping. The gray dashed lines are the reference data calculated with the conventional method implemented with the NESSi library [6]. (b) Change in the convergence error $\epsilon_{conv}$ for the real-time Green's function as a function of the number of iterations. The colored lines correspond to the colored regions shown in (a). We first iterate the self-consistent loop 100 times or until $\epsilon_{conv}$ reaches $1.0 \times 10^{-5}$, and then we change the cutoff from $10^{-12}$ to $10^{-16}$ and again iterate 100 times or until $\epsilon_{conv}$ reaches $1.0 \times 10^{-7}$.

data up to $t_{max} \approx 534$ on the scale of Figs. 14(a) and 14(b). However, in the zoomed-in plot of the order parameter $m$ in Fig. 14(c), a slight deviation from the reference data is observed for $t_{max} > 300$. Furthermore, the absolute errors of $n_\uparrow + n_\downarrow - 1$ and $E_{tot}(t) - E_{tot}(0)$ increase for $t_{max} > 300$ (see Figs. 14(d) and 14(e)). To obtain more accurate results at these longer timescales, it would thus be necessary to increase the number of iterations in the self-consistent loop for each block time stepping.

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

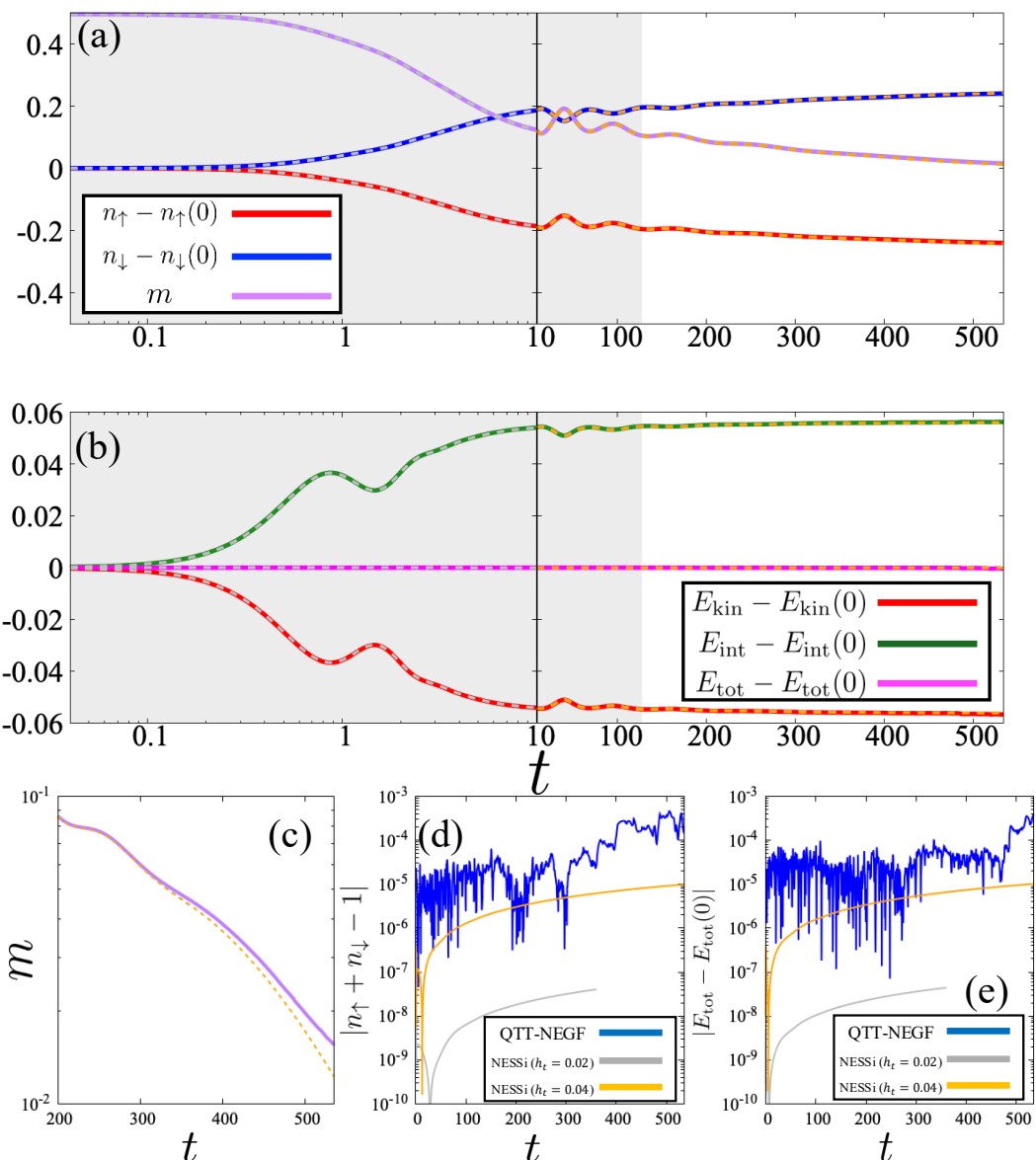

Figure 14: Time evolution of physical observables obtained with the divide-and-conquer algorithm for a quench from $U = 2$ to $U = 1.5$ in the AFM state of the Hubbard model up to longer times ($t_{\max} \sim 534$) than in the main text. (a) Time evolution of $n_{\uparrow}(t) - n_{\uparrow}(0)$, $n_{\downarrow}(t) - n_{\downarrow}(0)$, and the order parameter $m$. (b) Time evolution of $E_{\mathrm{kin}}(t) - E_{\mathrm{kin}}(0)$, $E_{\mathrm{int}}(t) - E_{\mathrm{int}}(0)$, and $E_{\mathrm{tot}}(t) - E_{\mathrm{tot}}(0)$. (c) Zoomed-in comparison of the order parameter $m$ between the QTT-NEGF method and the conventional method. (d) Absolute error of $n_{\uparrow} + n_{\downarrow} - 1$. (e) Absolute error of $E_{\mathrm{tot}}$ relative to the initial state $t = 0$. Gray and orange dashed or solid lines show results obtained with the conventional method using $h_t = 0.02$ and $h_t = 0.04$, respectively.