# Peer review of "A causality-based divide-and-conquer algorithm for nonequilibrium Green's function calculations with quantics tensor trains"

_SciPost Physics_

## Round 2 · Referee Report · Anonymous (Referee 1) · 2025-11-19

Strengths

1. This paper improves a previous method of finding non-equilibrium Green’s functions with Quantics Tensor Trains by using a causality-based divide and conquer algorithms, which in practice corresponds in solving the Green’s function globally for a given time $t_{max}$ and then fixing the Green’s function inside this time domain and increase the time domain slower $\Delta t$ and just update the part of the Green’s function corresponding to this $\Delta t$.
2. They use this method to find the non-equilibrium DMFT Green’s functions of the Hubbard model in the AFM phase and compare their results with the conventional approach implemented with NESSi.
3. They compare the data-size of the Green’s functions found by conventional methods and those found with QTT methods, finding an improvement of almost 3 orders of magnitude when compressing the data with QTT.

Weaknesses

  1. Although they estimate the runtime memory (or the number of operations) that needs the QTT method at each iteration (it scales as $\mathcal{O}(L D^3)$). They never discuss or estimate the approximate number of iterations that require traditional methods to achieve the final result. It would be interesting to know how those two methods compare in number of operations.

Report

In this work, the authors use a method based in Quantics Tensor Trains to simulate quench dynamics in symmetry broken phases with DMFT. This method improved the previous method published by some of the authors [M. Murray, H. Shinaoka and P. Werner (2024), arXiv:2412.14032] by using a causality-based divide-and-conquer algorithm they are able to avoid the “excessive number of iterations” required for the method to converge. Furthermore, they show results for the quench dynamics when the Hubbard interaction U is changed suddenly from U=2 to U=1.5 and show how the data size of the Green function’s calculated is between 2 or 3 orders of magnitude smaller with respect to conventional state-of-the-art methods (NESSi). The data-size does not seem to scale with $t_{max}$, although they report that for very long times, the energy and particle number conservation start to degrade.
All in all, I would recommend this paper for publication after a couple of minor issues are addressed.

Requested changes

Minor changes
1. The addition of a discussion between the number of operations required by conventional and the QTT block time stepping methods to find the Green’s functions. It does not need to be exhaustive I would be happy with a similar number with the one you give for the QTT block time stepping methods where you report the approximate number of operations required per iteration.
Very small changes in the manuscript
2. In the first paragraph of the introduction, you talk about the data-size and computational cost scaling with the total number of time steps, but in the end you say that it is difficult to simulate non-equilibrium dynamics in “large lattice systems and long times” without talking about the it is difficult to simulate large lattice systems or giving an idea on how the memory and operation cost scale with $N_x$.
3. In section (2.1) when you talk about the bond dimension typically you would like $D \ll 2^R$ so the data size is $\ll \mathcal{O}(4L2^{2R})$.

Recommendation

Ask for minor revision

  • validity: high
  • significance: good
  • originality: good
  • clarity: high
  • formatting: excellent
  • grammar: excellent

Author:  Ken Inayoshi  on 2025-12-08  [id 6118]

(in reply to Report 1 on 2025-11-19)
Disclosure of Generative AI use

The comment author discloses that the following generative AI tools have been used in the preparation of this comment:

GitHub Copilot in VS Code (ChatGPT-4.1) was used for spelling and grammar checking.

Category:
answer to question

We thank the referee for the careful review of our manuscript and for the valuable comments. Below, we provide detailed responses to these comments.

Comment 1:

The addition of a discussion between the number of operations required by conventional and the QTT block time stepping methods to find the Green’s functions. It does not need to be exhaustive I would be happy with a similar number with the one you give for the QTT block time stepping methods where you report the approximate number of operations required per iteration.

We thank the referee for pointing out this important aspect.

In the conventional method, the main computational cost per iteration arises from the convolution integral (matrix multiplication) in the Dyson equation. The runtime memory is proportional to the data size of the matrix representation of the Green's function, which scales as $\mathcal{O}(N_t^2)$, and the computational cost scales as $\mathcal{O}(N_t^3)$ (typically, $N_t$ is $10^3$ to $10^4$). In the QTT method, the main computational bottlenecks are solving the Dyson equation with a linear equation solver, element-wise products, and convolution integrals (the latter two represented as MPO-MPO contractions). The element-wise product is used for calculating self-energy diagrams (Eq. 24 in the main text), while the convolution integral is used in the calculation of constant terms $b$ in linear equations (Eqs. 16-19 in the main text). Storing the QTT representation of the Green's function requires $\mathcal{O}(D^2)$ memory for each component. However, the actual runtime memory scales as $\mathcal{O}(D^3)$ due to storing intermediate results of the MPO-MPO contractions. On the other hand, the computational cost scales as $\mathcal{O}(D^4)$.

Although the runtime memory and computational cost for a single QTT calculation (e.g., a single contraction) are moderate due to the small bond dimensions ($D=\mathcal{O}(10)$) in our calculations, our implementation involves multiple contraction operations for diagram calculations and the computation of the constant term $b$ appearing in the linear equation. Additionally, while the conventional method requires only a few iterations per time step ($h_t=0.02$) to achieve good accuracy, our approach, even with masking functions, requires several hundred iterations per block step ($t_{\mathrm{max}} \to t_{\mathrm{max}}+\Delta t, \Delta t =\mathcal{O}(10)$) as shown in the results.

As a result, the total computation time of our current implementation is longer than that for the conventional method. Specifically, for the first block time step ($t_{\mathrm{max}}=128\to 128\times 1.1\approx 141$), we need approximately 20-30 times longer than the NESSi calculation. However, at later times, the cost of the conventional time-stepping method increases, narrowing the performance gap. For the final block time step ($t_{\mathrm{max}}\approx 274\to 274\times 1.1\approx 301$), the QTT method is about 6 times slower than the NESSi calculation. Considering the total time extension from $t_{\mathrm{max}}=128$ to $t_{\mathrm{max}}\approx 301$, the QTT method is approximately 10 times longer than the NESSi calculation. Although our proof-of-concept QTT-NEGF implementation is currently slower than established implementations based on conventional methods such as the NESSi, the storage advantage is significant. In the current implementation, there is room for optimization in the tensor contractions and the linear equation solver. We expect that the computational time can be reduced by optimizing these components. Another promising direction is to extrapolate the initial guess for the self-consistent loop via dynamic mode decomposition. As demonstrated in our parallel work (arXiv:2509.22177), this approach can effectively reduce iteration counts, and we will address these improvements in future work.

In the revised manuscript, we will add this discussion at the end of Sec. 5.2.2, where the scaling of the runtime memory and computational cost is addressed. We will also extend the discussion about the future direction in the Conclusion section.

Comment 2:

In the first paragraph of the introduction, you talk about the data-size and computational cost scaling with the total number of time steps, but in the end you say that it is difficult to simulate non-equilibrium dynamics in “large lattice systems and long times” without talking about the it is difficult to simulate large lattice systems or giving an idea on how the memory and operation cost scale with $N_x$.

We thank the referee for pointing this out. For lattice systems, the computational and memory costs scale as $\mathcal{O}(N_{\boldsymbol{r}}^2)$, where $N_{\boldsymbol{r}}$ is the number of lattice sites. However, translational symmetry is often assumed in lattice systems, which simplifies the Green's function to being diagonal in momentum space. In this case, the computational and memory costs scale as $\mathcal{O}(N_k)$, where $N_k$ is the number of momentum points. Additionally, the Dyson equation can be solved independently for each momentum point, enabling efficient parallelization over momentum points. Still, the memory cost scales with the number of momentum points $N_k$, which is large in large lattice systems. As a result, simulating nonequilibrium dynamics in large lattice systems over long times remains a significant challenge.

At the end of the first paragraph of the introduction in the revised manuscript, we will add an explanation of how the memory cost scales with the number of momentum points $N_k$.

Comment 3:

In section (2.1) when you talk about the bond dimension typically you would like $D \ll 2^R$ so the data size is $\ll \mathcal{O}(4L2^{2R})$.

We thank the referee for the comment and apologize for not explicitly describing the data size of the original Green's function before compression. The data size of the original Green's function is $\mathcal{O}(N_t^2) = \mathcal{O}(2^{2R})$, where $N_t = 2^R$. When the bond dimension $D$ satisfies $D \ll 2^R$, the data size of the QTT representation, $\mathcal{O}(4RD^2)$, becomes exponentially smaller than that of the original representation.

In the revised manuscript, we will add this explanation in Sec. 2.1.

---

## Round 2 · Referee Report · Anonymous (Referee 2) · 2025-11-26

Strengths

1- feasible and useful extension of the QTT strategy for solving the KBE 2- Accurate benchmarks against time-stepping methods

Weaknesses

The paper lacks a discussion of the versatility of the QTT strategy in addressing more realistic Hamiltonians

Report

I have read with interest the paper entitled "A causality-based divide-and-conquer algorithm for nonequilibrium Green’s function calculations with quantics tensor trains" by Inayoshi et al.

The work presents an extremely useful advance in the quantics tensor trains (QTT) strategy for solving the Kadanoff–Baym equations. In essence, the authors have incorporated causality into the original QTT approach. This achievement allows them to extend the propagation time without the need to re-converge the Dyson equation over the entire extended domain.

The authors provide a thorough numerical analysis of convergence with respect to the number of iterations, successful benchmarks against the time-stepping method -- as implemented in the NESSI code -- and evidence that both particle number and energy are conserved during time propagation.

The paper is very well written, and I can recommend it for publication as is. The authors may, however, wish to include a discussion on scaling and memory requirements for simulating realistic systems (e.g., k-dependent Green’s functions and self-energies, multiple bands, long-range interactions etc.). Such a discussion would both broaden the scope of the QTT methodology and highlight the main challenges that must be addressed in order to make the KBE a competitive ab initio method. Related to this, the authors may also wish to discuss how QTT performs for other self-energies such as GW.

Requested changes

See report

Recommendation

Publish (surpasses expectations and criteria for this Journal; among top 10%)

  • validity: high
  • significance: high
  • originality: top
  • clarity: top
  • formatting: excellent
  • grammar: excellent

Author:  Ken Inayoshi  on 2025-12-08  [id 6119]

(in reply to Report 2 on 2025-11-26)
Disclosure of Generative AI use

The comment author discloses that the following generative AI tools have been used in the preparation of this comment:

GitHub Copilot in VS Code (ChatGPT-4.1) was used for spelling and grammar checking.

Category:
answer to question

We thank the referee for the thorough review and insightful comments.

As the referee mentioned, when simulating realistic lattice systems, we need to consider $k$- and band-dependent Green's functions.
In the current DMFT implementation, all MPS/MPO depend only on time variables, and we compress their time dependence.
Therefore, the memory and computational costs scale with the number of $k$-points, $\mathcal{O}(N_k)$, similar to conventional methods.

In our previous work (Phys. Rev. Lett. 135, 226501 (2025)), the QTT method (with only the linear equation solver) successfully handled large two-dimensional lattice systems with $\mathcal{O}(1000)$ lattice points, due to the high compressibility of the time dependence.
In that study, other self-energy diagrams (e.g., $GW$ approximation and Migdal approximation) were also calculated within the QTT framework.
Furthermore, in our recent work (arXiv:2509.22177), the divide-and-conquer algorithm combined with dynamic mode decomposition enabled long-time $GW$ simulations for large two-dimensional Hubbard models.

However, as the referee pointed out, for more realistic ab initio Hamiltonians, compressing real-space or momentum-space data is also necessary to reduce the memory cost.
The QTT framework can, in principle, be extended to compress both the time and momentum-space domains.
This is a promising direction for future work.

In the revised manuscript, we will add this discussion in the Conclusion section.

---

## Round 3 · List of Changes

We thank the editors for sending us the referee reports and two referees for their careful comments. Following the suggestions of the referees, we have made the following changes to the manuscript.
-
At the end of the first paragraph of the Introduction section, we have added an explanation of how the memory and computational costs scale with the number of momentum points.
-
In the final paragraph of Sec. 2.1, we have clarified the data size in the original representation of the Green’s function.
-
In the second to last paragraph of Sec. 5.2.2, we have included a discussion of the runtime memory and computational costs of both the conventional method and our QTT method. To consolidate the discussion, we moved the relevant text originally located in the paragraph starting with ”Figure 9 compares...” to this paragraph.
-
In the second to last paragraph of the Conclusion section, we have extended the discussion on preparing the initial guess of the Green’s function via the dynamic mode decomposition, as proposed in our parallel work (arXiv:2509.22177).
-
In the final paragraph of the Conclusion section, we have added a discussion of future directions for applying the QTT method to more realistic ab initio simulations.

---

## Editorial Decision

unknown